# Effect of Concomitant Proton Pump Inhibitors with Pazopanib on Cancer Patients: A Retrospective Analysis

**DOI:** 10.3390/cancers14194721

**Published:** 2022-09-28

**Authors:** Camille Moreau-Bachelard, Valentin Letailleur, Emmanuelle Bompas, Patrick Soulié, Julie Paul, Jean-Luc Raoul

**Affiliations:** 1Department of Medical Oncology, Institut de Cancérologie de l’Ouest, 44805 Saint-Herblain, France; 2Department of Medical Oncology, Institut de Cancérologie de l’Ouest, 49055 Angers, France; 3Department of Biostatistics, Institut de Cancérologie de l’Ouest, 44805 Saint-Herblain, France; 4Department of Clinical Research, Institut de Cancérologie de l’Ouest, 44805 Saint-Herblain, France

**Keywords:** proton pump inhibitors, cancer, tyrosine kinase inhibitors, pazopanib, drug-drug interactions, efficacy

## Abstract

**Simple Summary:**

Accumulated evidence shows that co-prescribing proton pump inhibitors (PPIs) with major anticancer drugs is frequently harmful. We conducted a retrospective analysis of cancer patients treated with pazopanib in our health center. In this cohort of 147 patients, both the efficacy and the toxicity of pazopanib decreased in patients taking concomitant PPIs.

**Abstract:**

The absorption of pazopanib depends on gastric pH. PPIs are frequently prescribed for cancer patients to modify gastric acidity, decreasing pazopanib absorption. The aim of our study was, retrospectively, to investigate the impact of PPIs on the clinical efficacy and safety of pazopanib in a cohort of patients treated in our health center. Of the 147 patients who were included retrospectively, 79 (54%) did not take PPIs concomitantly with pazopanib (cohort 1), while 68 (46%) patients did take PPIs concomitantly with pazopanib (cohort 2). The efficacy parameters were lower in patients taking pazopanib and PPIs: the i/tumor response was statistically different between the two cohorts (*p* = 0.008), in particular, with 19% vs. 3% of the objective response and 24% vs. 43% of progression in cohorts 1 and 2, respectively; ii/median overall survival was 17.6 (95% CI: 12.5–32.8) months in cohort 1 and 8.6 months (95% CI: 5.9–18.6) in cohort 2 (HR = 1.7 [95% CI: 1.2–2.5]; *p* < 0.006); on multivariable analysis, overall survival was associated with performance status, PPI intake, tumor location, hemoglobin, and PMN/lymphocyte ratio. In contrast, the dose reduction for toxicity and severe adverse events were (non-significantly) less frequent in cohort 1. To conclude, our study shows that combining PPIs with pazopanib has an adverse effect on overall survival. The clinical modifications that were observed are in line with a decrease in pazopanib absorption due to PPIs. This co-medication should be avoided.

## 1. Introduction

Pazopanib is an oral tyrosine kinase inhibitor (TKI) of vascular endothelial growth factor receptors (VEGFR-1, VEGFR-2, and VEGFR-3), platelet-derived growth factor (PDGFR-α and PDFGF-β), and c-kit, which is currently approved for advanced renal cell carcinoma (RCC) and advanced soft tissue sarcoma (STS) [1,2]. In aqueous media, pazopanib is very slightly soluble at pH 1.0 and is practically insoluble above pH 4.0 [3]. Proton pump inhibitors (PPIs) decrease the secretion of gastric acid; for example, omeprazole 20 mg taken once daily increases the median gastric acid pH from 1.7 to 4.6, while the median percentage of time at pH < 4 decreases from 89 to 35% [4]. PPIs are widely used, including in cancer patients, and are frequently prescribed by the patient’s GP or obtained off-label to alleviate heartburn or epigastric pain. In patients undergoing treatment for cancer, the prevalence of the TKI-PPI combination is estimated at 25% [5].

A study showed that the combined use of pazopanib and esomeprazole decreased the maximum concentration (Cmax) and the area under the curve (AUC) of pazopanib [6]. There is a strong relationship between systemic exposure to pazopanib and efficacy (assessed by the response rate and progression-free survival [PFS]) or safety [7,8]. The negative effect of combining pazopanib therapy with PPIs has been demonstrated in several retrospective studies. A pooled retrospective analysis, comparing patients treated for STS in two prospective trials with pazopanib or placebo, with or without the associated PPIs, showed a reduction in PFS and overall survival (OS) in patients who were concomitantly on PPIs and pazopanib (no effect was observed in the placebo arm), suggesting a detrimental role for this combination [9]. Another retrospective study from medical records, in a cohort of 91 patients treated with pazopanib for STS, demonstrated that the 42 patients taking concomitant acid-suppressive medications had shorter PFS and a trend toward less hypertension [10]. Conversely, two retrospective studies of patients on pazopanib for advanced RCC did not show any change in survival when pazopanib was combined with PPIs [11,12].

The negative impact of PPIs on the efficacy of TKIs (usually because of decreased bioavailability) and immunotherapies (through the modification of the microbiota) has been widely debated in recent months [13,14].

The aim of our study was to retrospectively investigate the impact of PPIs on the clinical efficacy and safety of pazopanib in a cohort of patients treated in our center.

## 2. Patients and Methods

### 2.1. Database

Patients from the Institut de Cancérologie de l’Ouest (ICO) (from both locations, Angers and Saint-Herblain) were included in the study. Patient-related data were collected from records, including patient demographics, pathology, and outcomes (PFS and OS), as well as treatment strategies, efficacy (response rate), and tolerance (according to NCI-CTC AE 4.0). Before using patients’ data, we checked, for each subject, that they did not object to the use of their personal data for medical research. In accordance with French regulations, the ICO is committed to following the MR-004 reference methodology of the CNIL (Commission Nationale de l’Informatique et des Libertés). Thereby, the project is registered on the public directory of studies under the MR of the Health Data Hub. The data processing was recorded in the “data processing register” made available to the CNIL by the ICO Data Protection Officer, under number 415. The protocol was approved by the Ethics Committee of Angers University (number 2022-071).

### 2.2. Study Population and Objectives

All patients aged 18 years or older and who received pazopanib for metastatic or locally advanced cancer between 01/2015 and 01/2021 were included in the study. Patients who received pazopanib as adjuvant therapy, who were not followed by our institution, and those who objected to the use of their data for research were excluded from the study. Our primary objective was to assess the impact of the concomitant use of PPIs and pazopanib on the drug’s efficacy and safety.

### 2.3. Statistical Analysis

Categorical variables were described using the number of people and the associated percentage. They were compared using a Pearson Chi-squared test or Fisher’s exact test, whenever appropriate. Quantitative variables were described using mean and standard deviation (SD). They were compared using Student’s *t*-test if the assumption of normality was met; otherwise, non-parametric statistics (median, extremes, quartiles) were made and compared using Wilcoxon’s test. The median follow-up of the population was estimated using the reverse Kaplan-Meier estimator. Survival times were calculated using the Kaplan-Meier method, and median survival times and rates with 95% confidence intervals (CI) were reported. OS was defined as the time from the date of initiation of treatment with pazopanib to the date of death (all causes) or the date of the last follow-up. PFS was defined as the time from the date of initiation of treatment with pazopanib to the date of progression or death or the date of the last progression-free follow-up. Progression was defined clinically or radiologically. For independent prognostic factors of event occurrence (death and progression) over time, univariable analyses were performed using Cox proportional hazards models, where the proportional hazards hypothesis was tested. Variables significant at the 15% level in univariable analyses were entered into a multivariable model, with a final significance level set at 5% (two-sided formulation). The strength of the association was estimated by the adjusted hazard ratio (HR), reported with a 95% CI. Missing data were described for each variable, but no imputation was performed. All analyses were performed with R software (R Core Team (2014). R: A language and environment for statistical computing. R Foundation for Statistical Computing, Vienna, Austria. URL: http://www.R-project.org/ accessed on 15 July 2022).

## 3. Results

### 3.1. Patient Characteristics

Of the 154 patients identified in the database, 147 met the inclusion criteria; 7 were excluded because of an absence of follow-up or a refusal to allow the use of their personal data. The median follow-up from diagnosis was 15.2 years [IQR = 12.3–NR]. Of the 147 patients, 79 (54%) did not take PPIs concomitantly with pazopanib (cohort 1), while 68 (46%) patients did take PPIs concomitantly with pazopanib (cohort 2). The median age at primary diagnosis was 65 years (IQR = 53–74). The cancer that was treated was RCC in 100 patients (68%), SFS in 36 patients (24%)m and “other” in 11 patients (7%). The cancer was metastatic for 138 patients (94%) and locally advanced for 9 (6%) patients. Pazopanib was given as a first-, second-, or third- or more line treatment for metastases for 54 (37%), 40 (27%), or 47 patients (32%), respectively. The median body mass index was 24.4 kg/m2 (IQR = 22.1–27.8). Patients were symptomatic, paucisymptomatic, or asymptomatic at pazopanib initiation in 55 (38%), 55 (38%), and 36 (24%) patients, respectively. No statistically significant difference was found between the two cohorts. The patients’ main clinical features are summarized in Table 1.

The PPIs used were esomeprazole, omeprazole, lansoprazole, pantoprazole, or rabeprazole in 29 (20%), 17 (12%), 14 (10%), 6 (4%) and 2 (1%) cases, all taken once daily. PPIs were started before pazopanib initiation and continued for more than 3 months in 40 (27%) cases, were begun shortly (<4 weeks) after pazopanib initiation and continued for more than 2 months in 8 (5%) cases, or were taken occasionally (more than 30% of the 3 first months of pazopanib) in 20 cases (14%). Major biological tests (hemoglobin, platelets, polymorphonuclear neutrophils (PMN), lymphocytes, calcemia, albuminemia, LDH, and platelet/lymphocyte and PMN/lymphocyte ratios) were in the range of normal values. Only the PMN values were significantly higher in the PPIs group when compared to the no-PPIs group (*p* < 0.012).

### 3.2. Impact of the Concomitant Use of PPIs and Pazopanib on Treatment Efficacy and Safety

The main features are summarized in Table 2. The median duration of pazopanib use was 4.4 [1.8–10.3] months in the overall population, 5.7 [2.5–12.8] months in cohort 1, and 3.8 [1.4–7.8] months in cohort 2 (*p* = 0.08). The initial dosage was the same in both cohorts (median 600 mg/d).

Tumor response was statistically different between the two cohorts (*p* = 0.008), particularly with 19% vs. 3% of objective responses and 24% and 43% of progression in cohorts 1 and 2, respectively.

Conversely, the number of patients reporting at least one side effect that was considered to be Grade 3 or more was 29% vs. 19%, a non-significant difference, but this was more frequent in patients without PPIs. We also observed more frequent dose reductions in pazopanib posology, due to toxicity, in cohort 1 (no PPIs) (58% vs. 38%) (*p* < 0.016).

The mOS was 17.6 (95% CI: 12.5–32.8) months in cohort 1 (no PPIs) and 8.6 months (95% CI: 5.9–18.6) months in cohort 2 (PPIs) (HR = 1.7 (95% CI: 1.2–2.5); *p* < 0.006)) (Figure 1).

The mPFS was 8.3 (95% CI: 5.7–14.0) months in cohort 1 (no PPIs) and 4.9 (95% CI: 3.3–10.2) months in cohort 2 (PPIs) [HR = 1.3 (95%CI: 0.9–1.9); *p* = 0.12] (Figure 1).

Of the 100 patients treated with pazopanib for renal cell carcinoma, 48 used PPIs. The mPFS was 11.3 months versus 9.2 months, HR = 1.27 [0.80–2.01], with *p* = 0.3 for those without and with PPIs, respectively. The mOS was 20.8 months versus 10.9 months, HR = 1.54 [0.97–2.45], *p* = 0.068 for those without and with PPIs, respectively.

If we extract those patients who took PPIs occasionally (*n* = 20) from the PPI cohort, an analysis focusing on patients with a continuous intake of PPIs showed that the mOS in this group was 8.4 (95% CI: 5.7–17.6) months, while the OS difference between the two cohorts (with or without PPIs) remained significant (HR= 1.9 [95% CI: 1.2–2.8]; *p* < 0.003); PFS in this group was 4.7 (95% CI: 3.1–12.6) months, with a trend in favor of cohort 1 (*p* = 0.088).

### 3.3. Univariable and Multivariable Analysis of the Parameters Associated with Survival

In the univariable analysis, biologicals, hemoglobin, platelets, PMN, lymphocytes, platelet/lymphocyte ratio, and PMN/lymphocyte ratio were associated with OS; the same parameters, plus LDH but without lymphocytes, were associated with PFS. Clinically, performance status, symptoms, tumor location, and PPI intake were associated with survival. The same figures were observed for PFS.

In terms of the multivariable analysis (Table 3), OS was associated with performance status, PPI intake, tumor location, hemoglobin, and PMN/lymphocyte ratio, while PFS was associated with performance status, PPI intake, tumor location, and the PMN/lymphocyte ratio.

## 4. Discussion

Pazopanib (Votrient^®^) is a TKI indicated in advanced RCC and STS at the posology of 800 mg/d, taken once daily [1,2,15]. For pazopanib, as well as for imatinib and sunitinib, exposure-outcome relationships have been established and therapeutic windows defined; it is now known that only between a half and a quarter of patients treated with all these three anticancer agents had drug levels within the predefined target range [16]. It is thus considered to be of the utmost importance that drug levels be measured throughout treatment [16]. A recent observational study showed, in 70 STS patients, that pazopanib concentration was independently associated with the risk of toxicity and of progression at 3 months (OR 4.21, 95% CI (1.47–12.12), *p =* 0.008) [17].

Drug-drug interactions are a major and under-recognized problem, particularly in cancer patients who frequently receive many different drugs. These interactions may not only increase toxicity but may also decrease efficacy. It is now well known that drugs that inhibit gastric acid secretion, particularly PPIs, may interact with many anticancer drugs that are given not only orally (TKI, CDK4/6 inhibitors, …) but also intravenously (pemetrexed, immune checkpoint inhibitors, …) [13,14,18,19,20,21,22]. It is well known that in the presence of PPIs, there is a reduction in the absorption of pazopanib [6,23].

The clinical consequences of these interactions between PPIs and pazopanib have been previously reported. A retrospective analysis of two prospective studies of pazopanib in STS (EORTC 62,043 and 62072) demonstrated that gastric-acid suppressive agents (GAS) concomitantly administrated (in 17.7% of the 333 eligible patients) was associated with shorter mPFS (2.8 vs. 4.6 months (HR, 1.49; 95% CI, 1.11–1.99; *p* = 0.01)), and mOS (8.0 vs. 12.6 months (HR, 1.81; 95% CI, 1.31–2.49; *p* < 0.01)); these effects were not observed in placebo-treated patients [9]. This was also shown in another series of patients treated for STS [10]. Surprisingly, two retrospective series analyzing pazopanib in advanced RCC patients gave negative results, with only minor differences in prognosis between patients receiving PPIs or not receiving them [11,12].

Our retrospective study reported our findings for a series of 147 patients, mainly treated for advanced RCC. Surprisingly, a large number of pazopanib patients received PPIs concomitantly (46.2%). This is partly related to the fact that we classified the 20 patients taking PPIs occasionally (more than one-third for the first three months of pazopanib treatment) in the PPI cohort. Both cohorts (with or without the concomitant use of PPIs) had similar clinico-biological characteristics. Despite a limited number of patients, this provides clinical evidence indicating that the concomitant use of PPIs with pazopanib is detrimental, with reduced survival and response rates in patients with advanced RCC or STS. The tumor objective response was statistically different between the two cohorts and was better in patients not taking PPIs (19% vs. 3%). Our results are consistent with less exposure to pazopanib when taken with PPIs: better tolerance (less severe adverse events (NS), less need for dose reductions (*p* = 0.016)), lower efficacy (lower tumor response (*p* = 0.008)), and poorer survival data (statistically significant and relevant for OS (17.6 vs. 8.6 months; *p* < 0.006), relevant but not statistically significant for PFS (8.3 vs. 4.9 months; *p* = 0.12)).

Pretreatment data from both cohorts (with and without PPIs) were very similar; only the PMN counts were significantly different. Of note, performance status, treatment line, and symptoms at pazopanib initiation were not greatly different between the two cohorts. This is of importance; it can, therefore, be considered that the difference in prognosis between these two groups may be related to more advanced disease with poorer performance status and more symptomatic patients, leading to the more frequent use of gastric acid-suppressive agents, particularly of PPIs. The independent prognostic value of PPI use was confirmed by multi-parametric analysis, highlighting the known poor prognosis factors, such as performance status, the symptoms and types of cancer, and certain biological parameters (hemoglobin level and PMN/lymphocyte ratio), showing exposure to PPIs at the same level of impact.

Most patients (100, 68%) in our cohort had advanced RCC, with STS representing less than a quarter of our population. Contrary to other series dedicated to RCC, we can, thus, hypothesize that the negative impact of concomitant PPI intake is not limited to STS and that mOS in RCC patients without PPIs is higher (non-significantly), being twice that observed in those who take gastric acid-suppressive drugs (20.8 months vs. 10.9 months; *p* = 0.068).

The major limitation of this study was its retrospective nature and the fact that the inclusion criteria were not limited to RCC or STS. We limited our analysis to those patients receiving PPIs as, currently, in France, histamine 2 receptor antagonists (cimetidine, ranitidine, famotidine, etc.) are no longer available. The importance of the proportion of patients receiving PPIs (46%) is surprisingly high. In a prospective cross-sectional study conducted in 2020 in four French cancer centers, including Angers and Saint-Herblain, we found that 26.3% of patients were taking PPIs; PPI use was most frequent among patients treated with TKIs (33.3%). In this series, only 38% of our patients had symptoms (as a whole), meaning that we can, thus, consider that PPI prescriptions may be given preventively. This may be related both to the fact that more than half of these patients were in second-line treatment and above (fragile), and to the toxicity profile of pazopanib with more frequent nausea, vomiting, and upper abdominal pain than with the placebo in adjuvant trials [24]. The number of patients that were included is also debatable, but despite this limitation, the negative impact of PPIs was clearly highlighted.

## 5. Conclusions

Our study, as with others, shows that combining PPIs with pazopanib has an adverse effect on the efficacy of the drug and decreases toxicity as a result of decreased drug exposure. The rationale is clear (poor pazopanib absorption due to the rise in gastric pH), with multivariable analysis confirming this relationship, and there is no clear bias in our cohort.

This co-medication between pazopanib and PPIs must be avoided.

## Figures and Tables

**Figure 1 cancers-14-04721-f001:**
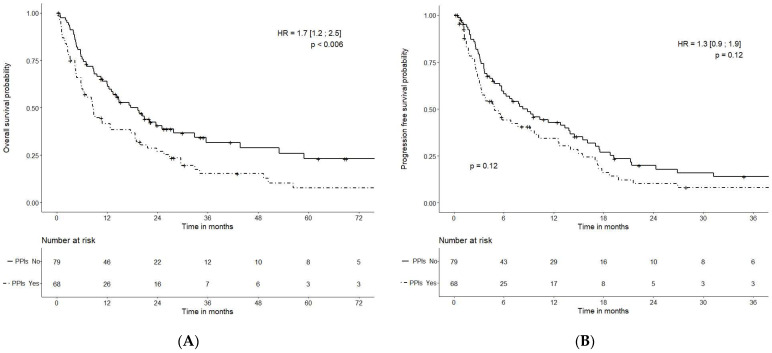
Overall survival (**A**) and progression-free survival (**B**), depending on PPI administration.

**Table 1 cancers-14-04721-t001:** Patient characteristics.

	Total (*n* = 147)	No PPIs (Cohort 1) (*n* = 79)	With PPIs (Cohort 2) (*n* = 68)	*p*-Value
Sex, *n* (%)				0.450
Women	60 (41%)	30 (38%)	30 (44%)	
Men	87 (59%)	49 (62%)	38 (56%)	
Age at metastatic diagnosis (years)				0.767
Mean (SD)	62.4 (14.9)	62.1 (14.9)	62.7 (14.9)	
Median [Min, Max]	65 (23–89)	64 (23–86)	66 (27–89)	
Body Mass Index				0.884
Mean (SD)	26.1 (10.78)	26.6 (13.7)	25.6 (5.5)	
Performance Status				0.962
0	23 (16%)	12 (15%)	11 (16%)	
1	87 (60%)	47 (60%)	40 (59%)	
2	31 (21%)	17 (22%)	14 (21%)	
3	5 (3%)	2 (3%)	3 (4%)	
Symptoms at pazopanib initiation				0.739
No	36 (24%)	21 (27%)	15 (22%)	
Yes	55 (38%)	27 (35%)	28 (41%)	
Pauci-symptomatic	55 (38%)	30 (38%)	25 (37%)	
Location of primary cancer, *n* (%)				0.149
Renal	100 (68%)	52 (66%)	48 (71%)	
Soft-tissue sarcoma	36 (24%)	18 (23%)	18 (26%)	
Other	11 (7%)	9 (11%)	2 (3%)	
Treatment line, *n* (%)				0.770
Locally advanced	6 (4%)	4 (5%)	2 (3%)	
Metastatic, first line	54 (37%)	31 (39%)	23 (34%)	
Metastatic, second line	40 (27%)	21 (27%)	19 (28%)	
Metastatic, ≥third line	47 (32%)	23 (29%)	24 (35%)	
Duration of treatment (months)				
Mean (SD)	8.2 (10.2)	9.3 (10.8)	7.0 (9.4)	0.086
Median [Min, Max]	4.4 [0.1; 55.0]	5.7 [0.1; 52.7]	3.8 [0.3; 55.0]	

PPIs: proton pump inhibitors; SD: standard deviation; [Min, Max]: [Minimum, Maximum].

**Table 2 cancers-14-04721-t002:** Summary of the safety and efficacy data for pazopanib, following the concomitant intake of PPIs.

	Total (*n* = 147)	No PPIs (Cohort 1) (*n* = 79)	With PPIs (Cohort 2) (*n* = 68)	HR [95% CI]	*p*-Value
Status at last follow-up (May 2022) Alive Death	39 108	28 51	11 57		0.008

Overall survival, median (CI95%)	12.7 mo (9.1–19.3)	17.6 mo (12.5–32.8)	8.6 mo (5.9–18.6)	1.7 [1.2–2.5]	<0.006
Progression-free survival, median (CI95%)	6.8 mo (4.9–10.3)	8.3 mo (5.7–14.0)	4.9 mo (3.3–10.2)	1.3 [0.9–1.9]	0.12
Tumoral response (RECIST), *n* (%)					0.008
Objective response	15 (12%)	13 (19%)	2 (3%)		
Stable disease	72 (56%)	40 (57%)	32 (54%)		
Progressive disease	42 (32%)	17 (24%)	25 (43%)		
Missing data	18	9	9		
Serious toxicity, per patient, *n* (%)					0.160
No	111 (76%)	56 (71%)	55 (81%)		
Yes	36 (24%)	23 (29%)	13 (19%)		
Reduction in dosage following severe toxicity, per patient *n* (%)					0.016
No	75 (51%)	33 (42%)	42 (62%)		
Yes	72 (49%)	46 (58%)	26 (38%)		

PPIs: proton pump inhibitors; HR: hazard ratio; CI: confidence interval; RECIST: response criteria in solid tumors.

**Table 3 cancers-14-04721-t003:** Multivariable analysis for those factors associated with overall survival.

Parameter	HR	95%CI	*p*-Value
Performance status (ref = 0–1)	2.78	[1.77–4.38]	<0.0001
PPI intake (ref = no)	2.00	[1.34–2.99]	0.0008
Tumor location (ref: RCC)	2.51	[1.62–3.89]	<0.0001
Hemoglobin (g/dL)	0.90	[0.82−0.99]	0.0303
PMN/lymphocyte ratio	1.13	[1.08–1.19]	<0.0001

HR: hazard ratio; CI: confidence interval; PPIs: proton pump inhibitors; RCC: renal cell cancer; PMN: polymorphonuclear neutrophils.

## Data Availability

Data access may be provided on demand to the Correponding Author.

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
