# Peer review of "Effect of Concomitant Proton Pump Inhibitors with Pazopanib on Cancer Patients: A Retrospective Analysis"

_cancers, 2022, doi:10.3390/cancers14194721_

Round 1
Reviewer 1 Report
In this study, the authors performed a retrospective study on the concomitant usage of PPI and RTK inhibitor pazopanib in a single clinical center in France. They found out after careful analyses that co-administration of PPI drug and pazopanib is detrimental and strongly advise the concomitant usage to be avoided.
While this conclusion is significant in the clinic, it does not add novel knowledge to the scientific field. Basically it is a confirmative study for something already known by the community.
Putting novelty shortage aside, this retrospective study is nicely designed and conducted.
Author Response
We would like to thank reviewer #1 for his very positive comments and like him, we expect that repeating the message is positive.
Reviewer 2 Report
The manuscript presented a study on a retrospective analysis of effect of concomitant proton pump inhibitors with 2 pazopanib on cancer patients. The authors used various biostatistics method to prove their conclusions that taking concomitant proton pump inhibitors has an adverse effect on overall survival. The reviewer recommends publications of the manuscript after the authors answer the following question:
In Table 1 patient characteristics, duration of treatment, we can see that no PPIs (cohort1) is shorter than With PPIs (cohort 2). e.g. 9.3 vs 7.0, 5.7 vs 3.8, although p value >0.05, but that may weaken the author's conclusion because longer treatment time mean longer survival time for the patients. Can the author further explain this such as why with PPIs (cohort 2) have shorter treatment time because treatments with PPIs should have less toxicity and should have longer treatment time due to better drug tolerance for patients.
Author Response
We would like to thank reviewer #2 for his positive comments.
Regarding data about duration of treatment the reviewer misunderstood our Table : in fact duration of treatment was longer in patients without PPIs (mean and median) certainly in relation with a better therapeutic efficacy.
In order to limit the risk of mistake we have modified the sentences adding with or without PPIs after the name of the cohort.
Round 2
Reviewer 1 Report
The manuscript has been sufficiently improved to warrant publication in Cancers